# Lifespan extension in female mice by early, transient exposure to adult female olfactory cues

**Michael Garratt[1]\*, Ilkim Erturk[2], Roxann Alonzo[2], Frank Zufall[3], Trese Leinders-Zufall[3], Scott D Pletcher[4], Richard A Miller[2]**

[1]Department of Anatomy, School of Biomedical Sciences, University of Otago, Dunedin, New Zealand; [2]Department of Pathology and Geriatrics Center, University of Michigan, Ann Arbor, United States; [3]Center for Integrative Physiology and Molecular Medicine, Saarland University, Homburg, Germany; [4]Department of Molecular and Integrative Physiology, University of Michigan, Ann Arbor, United States

**\*For correspondence:**
mike.garratt@otago.ac.nz

**Competing interest:** The authors declare that no competing interests exist.

**Abstract** Several previous lines of research have suggested, indirectly, that mouse lifespan is particularly susceptible to endocrine or nutritional signals in the first few weeks of life, as tested by manipulations of litter size, growth hormone levels, or mutations with effects specifically on early-life growth rate. The pace of early development in mice can also be influenced by exposure of nursing and weanling mice to olfactory cues. In particular, odors of same-sex adult mice can in some circumstances delay maturation. We hypothesized that olfactory information might also have a sex-specific effect on lifespan, and we show here that the lifespan of female mice can be increased significantly by odors from adult females administered transiently, that is from 3 days until 60 days of age. Female lifespan was not modified by male odors, nor was male lifespan susceptible to odors from adults of either sex. Conditional deletion of the G protein Gαo in the olfactory system, which leads to impaired accessory olfactory system function and blunted reproductive priming responses to male odors in females, did not modify the effect of female odors on female lifespan. Our data provide support for the idea that very young mice are susceptible to influences that can have long-lasting effects on health maintenance in later life, and provide a potential example of lifespan extension by olfactory cues in mice.

## Editor's evaluation

This valuable study provides solid evidence for a new intervention, exposure to male vs. female olfactory cues, with an impact on female mouse lifespan. This is interesting to the field of aging research, especially since most described pro-longevity interventions to date tend to work better in male mice.

## Introduction

Interventions in the first few weeks of postnatal life seem to have particular power to produce life-long changes in mouse physiology, with effects on late-life illnesses and on survival. Bartke's laboratory, for example, has shown that the exceptionally long lifespan of the Ames dwarf mice, in which longevity appears to reflect lower levels of growth hormone (GH) produced by the anterior pituitary, can be reduced back to that of non-mutant controls by transient exposure to daily GH injections started at 2 weeks of age and discontinued 6 weeks later (*Panici et al., 2010*). Furthermore, these early-life GH

**eLife digest** The environment that animals are exposed to early in life can influence their subsequent rate of development, reproduction and aging. Experiments done in rodents have shown that social stimuli such as odours from the same sex or opposite sex individuals can affect the age at which sexual maturity is reached. Variations in age of sexual maturity are directly correlated with median lifespans of mice, with strong associations observed between later sexual maturity and longer lifespans in female mice.

Detailed experiments exposing female or male mice to scents from mice of the same or another sex strongly suggest that growing up smelling the same sex can delay sexual maturity, while scents from another sex can hasten it. Interestingly, mice that lacked the cells that sense odours do not change their age of sexual maturity in response to scents from the opposite sex. This ability to steer one's developmental timeline depending on environmental cues may allow animals to prepare for future environments. But can it also influence an animal's lifespan?

To answer this question, Garratt et al. observed the lifespans of female and male mice under different conditions. Mice were exposed to same-sex or other-sex odours, in the form of urine or soiled bedding, from day 3 to day 60 of their lives. The results showed that female mice exposed to odours from other females exhibited an increased lifespan, as compared to those not exposed to scents, while those exposed to odours from males did not show any change in their lifespan. In striking contrast, male mice exposed to odours from either sex showed no variation in their lifespans. The impairment of a particular type of odour-sensing neuron in mice did not change these results, making it likely that another neuron type is responsible for the changes in lifespan observed in the female mice.

These experiments elegantly demonstrate that exposure to certain sensory information, in this case scent, can change how long mammals live. While similar effects involving smells are unlikely to influence lifespan in humans, it is possible that other types of sensory information affect our health and how we age.

injections block the development of multi-modal stress resistance shown by skin-derived fibroblasts of Ames dwarf mice (*Panici et al., 2010*), and also prevent the lower levels of hypothalamic inflammatory characteristics of unmanipulated Ames mice (*Sadagurski et al., 2015b*). Many other characteristics of Ames dwarf mice, plausibly connected to their longevity, insulin sensitivity (*Dominici et al., 2002*), and cognitive health (*Kinney et al., 2001*; *Sun et al., 2005*), are also blocked by early-life GH treatments when tested at 18 months of age, including changes in fat, muscle, plasma irisin and GPLD1, liver, and brain (*Li et al., 2022*). Conversely, lifespan can be increased in genetically normal mice by pre-weaning manipulation of litter size – mice weaned from litters in which litter size has been increased to 12 pups/nursing mother live longer than those in control litters of 8 pups (*Sun et al., 2009*). Genetic selection experiments have also provided relevant data, by showing extended lifespan in non-inbred stocks of mice that were selected, for 17 or more generations, for exceptionally slow growth in the first 10 days of life, or between 26 and 56 days of age (*Miller et al., 2000*).

Age at sexual maturity is another early-life developmental factor that has been associated with lifespan. Across strains of mice, age at sexual maturity correlates positively with median lifespan (*Yuan et al., 2012*), and genetic manipulations that delay sexual maturity have been associated with longer lifespan in females (*Wang et al., 2018*). Age at sexual maturity in rodents is also influenced by environmental cues, in particular olfactory cues from conspecifics. Female mice that are exposed to male odors (typically soiled bedding or male urine) during development show hastened sexual maturity (*Drickamer, 1983*; *Vandenbergh, 1973*). In contrast, exposing females to the odors of group-housed adult virgin females can delay sexual maturity (*Vandenbergh, 1973*; *Drickamer, 1977*). In males, exposure to females or their odors has been shown to be associated with increased masses of reproductive organs (testes, seminal vesicles) early in life, also indicating earlier sexual maturity in males (*Vandenbergh, 1971*). These odor priming effects are assumed to be adaptive, allowing animals to modulate their timing of early-life reproduction to suit predicted future environments. In one previous study, hastened female sexual maturity was associated with an increase in mortality over the first 180 days of life and reduced litter sizes when breeding, suggesting a cost to this developmental

change (*Drickamer, 1988*). However, in this prior study females were exposed directly to males on the day of their first estrous, which was earlier in females exposed to male odors producing a confounding effect that could contribute to mortality differences.

The effects of odors on sexual maturity are initiated, at least partly, by detection of odorants through the vomeronasal organ, one of the two olfactory sub-systems in mice. Female mice that have had their vomeronasal organ surgically removed do not show hastened sexual maturity in response to male odors (*Lomas and Keverne, 1982*). Vomeronasal sensory neurons (VSNs) express hundreds of different sensory receptors, but these receptors largely use one of two different types of G protein for signal transduction (*Munger et al., 2009*). The G protein Gαo is used for signal transduction in VSNs of the basal layer of the vomeronasal organ. Cre-mediated deletion of Gαo in the olfactory system has been shown to inhibit females from showing hastened sexual maturity in response to male odors (*Oboti et al., 2014*), indicating that this subset of VSNs is important in driving changes in sexual maturity, at least in response to opposite sex cues in females.

With these precedents in mind, we tested the idea that olfactory signals, presented to mice from 3 days of age but discontinued at 60 days, would have long-lasting effects that influence age at death. We initiated odor exposure early in life, prior to weaning, because a previous study had shown that very early exposure influences female body weight during development (*Cowley and Wise, 1972*). In particular, we speculated that odor from adult female mice would lead to lifespan extension in female (but not in male) mice, and vice versa. We also hypothesized that exposure to opposite-sex odors would reduce survival given the hastening of sexual maturity. Here, we show that transient exposure to odor from adult females starting at 3 days of age can extend lifespan of female mice.

## Results

### Survival

Our principal hypothesis was that early-life exposure to sex-specific olfactory cues indicative of social environment would influence lifespan of mice. A secondary hypothesis was that the response to those cues would depend on function of Gαo-expressing neurons in the vomeronasal organ, since these have been shown to mediate early-life priming responses to opposite-sex urine, at least in female mice (*Oboti et al., 2014*). Mice were exposed either to same-sex or to opposite-sex odors starting from 3 days of age, with daily exposure through day 60 of life. They were housed in same-sex cages from the age of weaning, that is from 19 or 20 days. *Figure 1* shows survival curves for female and for male mice subjected either to odors from adult males ('MU') or from adult females ('FU'), or exposed to water as a control (ZU, for zero urine). Data from male and female mice were evaluated separately. *Table 1* shows median age, and age at 90th percentile, for each combination of sex and treatment, pooling across genotypes. In female mice, FU led to an 8% increase in median lifespan, compared to the ZU control group, and a 9% increase in the age at 90th percentile. The log-rank test, comparing all three treatment groups, showed that treatment led to significant differences among the groups (p=0.04), and a follow-up test showed that for females the FU mice were longer-lived than the ZU controls (p=0.01 by log-rank test). Male odors did not produce a lifespan change in female mice compared to controls (p=0.6). Treatment did not modify lifespan in the male mice; log-rank p=0.8 for all treatment groups taken together.

To see if the effect of odor was dependent on Gαo expression in the vomeronasal organ, survival data were analyzed by Cox regression, with two factors: treatment (MU, FU, or ZU) and genotype (WT for wild-type [*Gnao1*fx/fx, *Omp*+/+] and mutant for *Omp*cre positive mice [*Gnao1*fx/fx, *Omp*cre/+]), with a [treatment × genotype] interaction term. A significant interaction would indicate that the two genotypes of mice have responded differently to the odor treatment. *Table 2* shows the significance levels for each of the three terms, calculated separately for female and for male mice. The interaction term did not achieve significance in either sex, implying that the odor effect did not depend on genotype. In addition, there was no independent effect of genotype on survival. Only the odor treatment had a significant effect on survival, and only in female mice (p=0.04), consistent with the results of the log-rank tests. The Cox regression calculation also confirmed the inference that FU females lived longer than ZU females (p=0.015 without adjustment for multiple comparisons, and p=0.045 with Sidak adjustment), and that no other pairwise comparison between treatment groups had a significant effect in either sex. Within each genotype and sex individual comparison (e.g. WT females, mutant

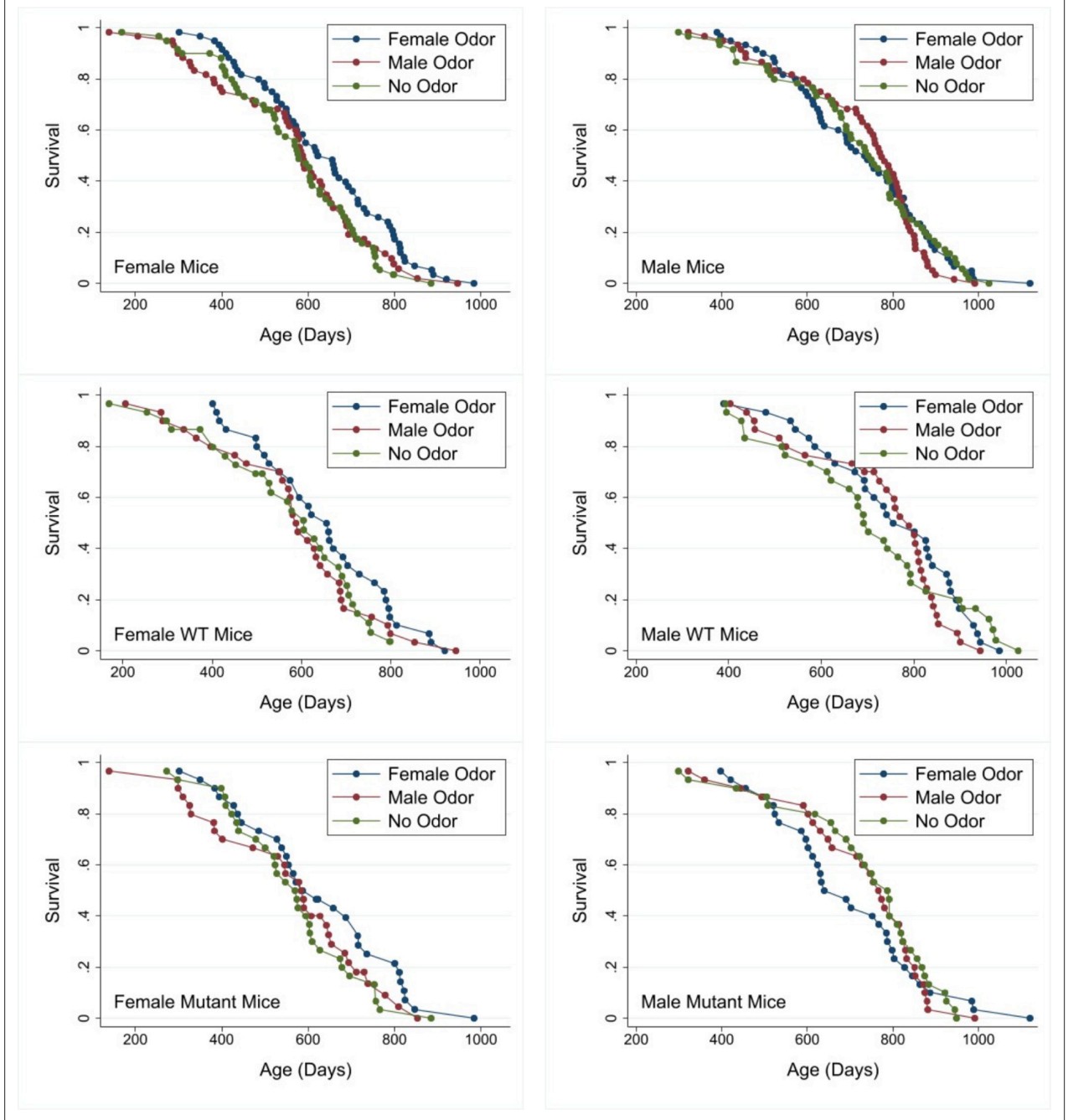

**Figure 1.** Survival curves for female (left panels) and male mice (right panels) exposed to male, female, or no odors daily from 2 to 60 days of age. Top row: pooled across genotypes. Middle row: wild-type (WT) mice. Bottom row: mutant mice. Cox regression analysis (see *Table 2*) shows no significant interaction between genotype and odor treatment in either sex. See the Supplementary data for age at death of each mouse and its genotype, sex, and treatment group.

The online version of this article includes the following source data for figure 1:

**Source data 1.** Individual-level survival data in days.

females, WT males, mutant males), there was no significant difference in survival between odor treatment groups using a log-rank test. This further emphasizes the lack of any strong genotype-specific effect and that the effects of the odor treatment are more clearly apparent when both genotypes are included within an analysis due to the increased sample size.

**Table 1.** Median and 90th percentile survival statistics for mice treated with same-sex or opposite-sex odors from 3 to 60 days of age.

Female mice, pooled across genotype

| Treatment | Count | Median | % Change | p(90) | % Change |
|---|---|---|---|---|---|
| Female odor (FU) | 59 (30 WT) | 621 | 8 | 823 | 9 |
| Male odor (MU) | 58 (30 WT) | 585 | 2 | 792 | 5 |
| No odor (ZU) | 57 (27 WT) | 576 | (NA) | 754 | (NA) |

Male mice, pooled across genotype

| Treatment | Count | Median | % Change | p(90) | % Change |
|---|---|---|---|---|---|
| Female odor (FU) | 60 (30 WT) | 737 | −1 | 932 | −1 |
| Male odor (MU) | 59 (29 WT) | 772 | 4 | 879 | −7 |
| No odor (ZU) | 59 (29 WT) | 741 | (NA) | 944 | (NA) |

To evaluate the effects of early-life odor exposure on survival to unusually high ages, we used the method of Wang and Allison (*Wang et al., 2004*), which calculates a Fisher's exact test statistic on the proportion of mice in each treatment that remain alive at the 90th percentile of their joint survival distribution, the method also employed by the mouse Intervention Testing Program consortium (*Macchiarini, 2021*). In females, the comparison of FU to control ZU mice produced a significance value of p=0.004, showing that FU-treated females were more likely to reach old age than ZU-treated females. None of the other paired comparisons reached significance in female mice, and none did so in male mice.

## Functional testing

Each mouse surviving to 22 months of age was evaluated using a series of tests for age-sensitive physiological function. The data were analyzed separately for each sex, using a two-factor ANOVA (treatment, genotype, interaction). Because there was no significant interaction between odor treatment and genotype for any measured trait, the data were pooled across genotypes for analysis of odor effects. *Figure 2* (top row) shows core body temperature at 22 months. For female mice, exposure to female odors leads to a significantly higher body temperature compared to the ZU control group (p=0.007). There are no significant differences between the MU group and FU mice or ZU controls, and there are no significant differences among any of the groups for male mice. *Figure 2* also shows (bottom row) levels of non-fasting glucose measured at 12 months of age. There were no effects of treatment in female mice, but glucose levels were significantly lower in male ZU mice than in the FU or MU mice (*Figure 2c*; p=0.004 for ANOVA, and p=0.01 for each pairwise comparison to ZU male mice). There were no treatment effects, or significant interaction terms, for mean forepaw grip strength or for rotarod performance (mean of three trials) in either sex (*Figure 2—figure supplement 1*).

*Figure 3* shows weight trajectories for each of the odor treatment groups separately in both sexes. There were no significant differences at any age, except that in female mice, ZU females were marginally heavier than mice in the MU or ZU control groups at 6 months of age; p=0.03, and p=0.04 if an MU mouse weighing 16 g is considered an outlier. Because these significance tests were not adjusted for multiple comparisons and because there were no large or significant effects in odor at any other age, we suspect that this difference at 6 months is a chance effect. Body weight changes with odor

**Table 2.** Significance tests for predictors in Cox regression, calculated separately for each sex.

| Sex of mouse | Treatment | Genotype | Interaction |
|---|---|---|---|
| Female | $X^2 = 6.5$ p=0.039 | $X^2 = 0.4$ p=0.5 | $X^2 = 0.12$ p=0.9 |
| Male | $X^2 = 0.52$ p=0.8 | $X^2 = 0.23$ p=0.6 | $X^2 = 0.31$ p=0.5 |

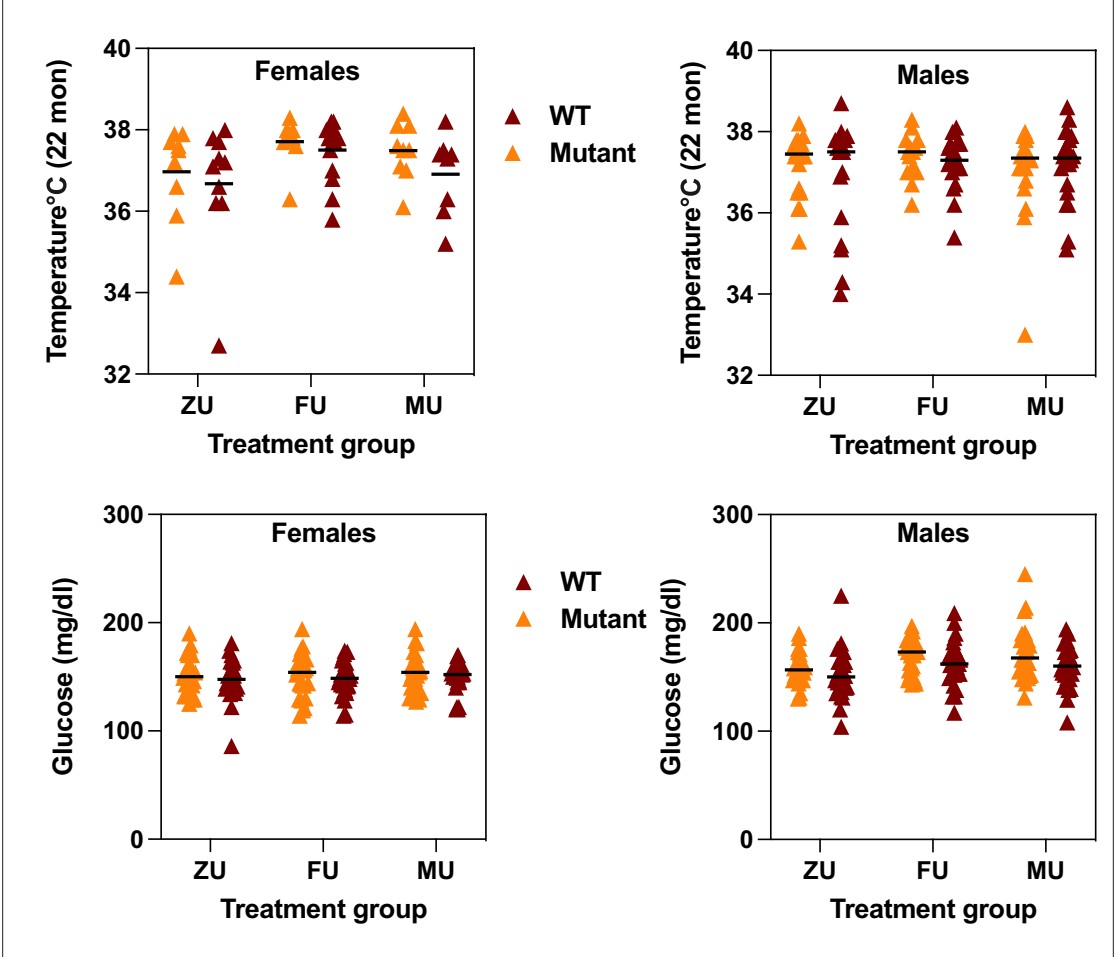

**Figure 2.** Measurement of core body temperature at 22 months (top row) and of plasma non-fasting glucose at 12 months (bottom row) in female (left) and male (right) mice of the indicated odor treatment groups. Each symbol represents one mouse, with color indicating genotype. The black line indicates the mean of each group. ZU = control mice, FU = those exposed to female odors, MU those that were exposed to male odors. Full data matched to each genotype and treatment group is available in the Supplementary data.

The online version of this article includes the following source data and figure supplement(s) for figure 2:

**Source data 1.** Individual data for temperature, grip strength, and rotarod function.

**Source data 2.** Individual data for glucose levels.

**Figure supplement 1.** No effect of genotype or odor treatment on grip strength or rotarod balance capacity.

treatment were also uninfluenced by genotype, and the genotypes of mice had similar body weights across the experiment (*Figure 3—figure supplement 1*).

## Discussion

Our results add to the growing evidence that late life health, including risks of lethal diseases, can be influenced by non-genetic factors, such as litter size (*Sun et al., 2009*; *Sadagurski et al., 2015a*) and GH levels (*Panici et al., 2010*; *Sadagurski et al., 2015b*), to which a newborn or pre-weanling mouse is exposed transiently. The 'window of opportunity', or age at which a mouse is susceptible to such influences, is not known, and may differ depending on the specific form of environmental variation, but it is noteworthy that GH treatments can prevent lifespan extension of Ames dwarf mice when these are initiated at 2 weeks of age, but fail in Snell dwarf (*Vergara et al., 2004*) and Ames dwarf animals (Bartke, unpublished results) when started at 4 weeks of age. The basis for this transient susceptibility to factors that set the tempo of late-life decline is unknown, but seems likely to involve

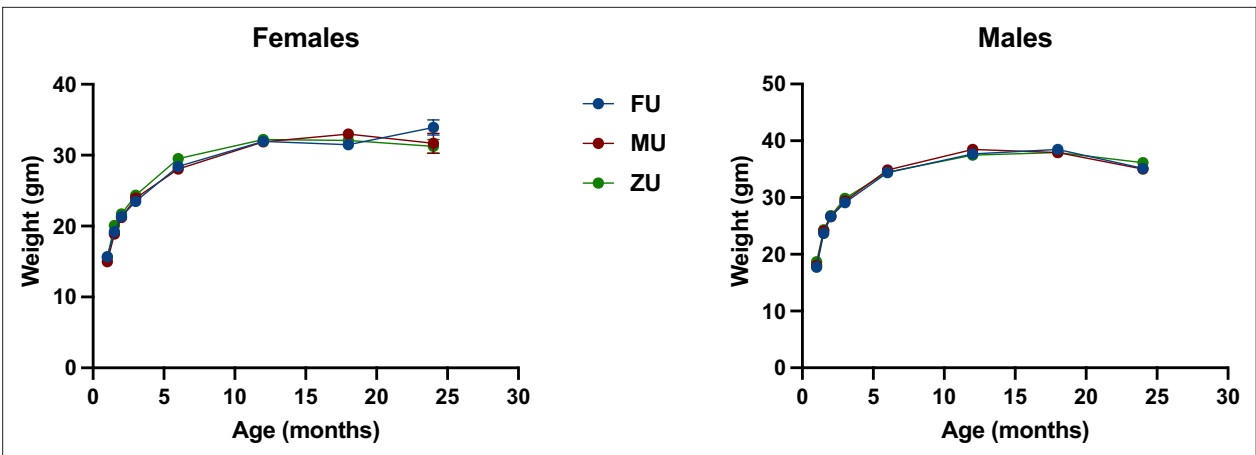

**Figure 3.** Weight as a function of age in female and male mice. The genotypes are pooled in the figure with separate figures provided in *Figure 3—figure supplement 1*. N=60 in each group through month 3, and then diminished to 32–39 mice (females) or to 48–49 mice (males) by month 18. Standard error of the mean (SEM) values are plotted, but are in most cases too small to see; SEM was <0.3 g at younger ages and typically 0.7–1.3 g at 18 months.

The online version of this article includes the following source data and figure supplement(s) for figure 3:

**Source data 1.** Individual level data for body weight at each time point.

**Figure supplement 1.** Weight as a function of age in female and male mice stratified by genotype.

epigenetic modulation of events that are sensitive to nutritional and neuro-endocrine stimuli. Fibroblast cultures from adult mice of the Ames and Snell dwarf mutant strains are resistant, in culture, to multiple forms of lethal injury (*Murakami et al., 2003*; *Salmon et al., 2005*; *Salmon et al., 2008*), unless the Ames mice had been subjected to early-life GH treatment (*Panici et al., 2010*). In this context it is noteworthy that fibroblasts derived from 1-week-old Snell dwarf and control mice do not differ in stress resistance (*Salmon et al., 2005*) at this age GH levels are still largely derived from the intra-uterine blood circulation, and are not yet controlled by the pituitary of the newborn mouse itself. GH exposure experience in the 2- to 8-week-old mouse (*Sadagurski et al., 2015b*), as well as milk availability mediated by the size of the nursing litter (*Sadagurski et al., 2015a*), also modulates the hypothalamic inflammatory state in mice 18 months of age or older. Ames mice treated transiently with GH also resemble WT, that is, non-mutant, controls in adipose tissue UCP1 levels, relative ratios of pro-inflammatory and anti-inflammatory macrophages in adipose tissue, production of FNDC5 and its myokine cleavage product irisin by skeletal muscle, hepatic and plasma levels of GPLD1, and brain levels of protective factors BDNF and DCX (*Li et al., 2022*), showing that these early-life environmental factors can have long-lasting, presumably permanent, effects on many physiological properties of high relevance to health maintenance and disease resistance.

The data reported here document another vector for early-life modulation of late-life fitness and mortality risks, by manipulation of conspecific social odors. We chose to study this topic because of data showing effects of early exposure to same-sex and opposite-sex odors on the pace of mouse development and sexual maturation. We hypothesized that exposure to same-sex odor, which delays development in some circumstances (*Drickamer, 1988*; *Drickamer, 1982*), might also delay the rate of age-related decline, and we found that indeed female mice exposed early to odors from adult females had longer lifespan than females exposed to water. Male odor did not have this effect on female longevity. The FU-treated female mice also resisted the age-related decline in core body temperature, although there was no inter-group difference in grip strength or rotarod performance, other indices of late-life physiological state. Male mice, in contrast, did not exhibit lifespan or performance effects in response to either same-sex or opposite-sex odors. We do not know the mechanism for the lifespan effect in the FU-treated female mice, or why we saw no such effect in male animals; these are questions we plan to explore in future work.

We should note that the genotype of the mice exposed to odor was not the same as the genotype (UM-HET3 genetically heterogeneous stock) from which urine and bedding was collected. The use of a genetically heterogeneous mouse model as odor donors would mean that the odor receiving animals

would be exposed to signaling proteins and volatile odorants produced by several strains of mice. We perceived this as a strength because different mouse strains produce different olfactory signaling proteins (*Cheetham et al., 2009*), and exposing mice to odors from a variety of genotypes would increase the probability of exposure to an odorant that influences lifespan. However, different mouse strains also express different olfactory receptors (*Ibarra-Soria et al., 2017*), and could show different physiological responses to bedding from different strains, although VNO activation responses in mice to odors from the same or different strain of mouse has been reported as broadly similar (*Bansal et al., 2021*; *Silvotti et al., 2018*). In principle, it's possible that different lifespan responses may occur if mice are exposed to odors from their background genotype.

The effects of odor exposure on lifespan were not modified by deletion of Gαo in the olfactory system. This G protein is important in mediating responses to peptide and protein-based ligands in basal VSNs (*Chamero et al., 2011*), and has previously been shown to inhibit priming responses to male odors in females, including changes in sexual maturity (*Oboti et al., 2014*). We therefore designed this experiment using this model because this deletion is known to inhibit priming effects, but does not affect normal body development or suckling behavior as has been observed with manipulations that impair main olfactory system function (*Mandiyan et al., 2005*). Previously published research with this model has largely focused on responses to male odors, but in this study we observed changes in lifespan in response to female odors. Exposure of female mice to female odors has previously been shown to activate a different subset of VSNs compared to male odorants, including apical VSNs that use a different G protein in signal transduction (*Silvotti et al., 2018*). Thus, effects of female odors on female survival may be mediated by a subset of VSNs different from those that were inhibited in this study. It is also possible that effects could be mediated by detection of odorants by the main olfactory system. As our results were not significantly altered by the olfactory genetic manipulation employed in this experiment, it is possible that a different sensory modality might be involved in mediating these effects. Similarly, we cannot exclude the possibility that microbiota transferred with the bedding, rather than the urine odor and bedding odor per se, cause these effects. Regardless, this must be a female-specific secreted factor that only influences female lifespan, and can do so even when females are exposed only over the first 60 days of life.

Changes in sexual maturity with exposure to conspecific odors can be initiated even by brief exposure to the diluted urine of a conspecific after weaning, although longer-term effects on female body weight have been observed when odor exposure begins prior to weaning (*Cowley and Wise, 1972*), which is why we chose an extended treatment protocol to maximize the chance of detecting effects on survival. Future studies with a shorter treatment time course after weaning, and with only urine or specific urinary fractions, would help to reveal the signals and developmental periods that are important causing subsequent changes in survival. Use of a different genetic model of impaired VNO signaling, for example disruption of the apical layer of the VNO, or a main olfactory system deficit mouse model, would help to firmly establish the role of the olfactory systems in mediating these effects.

Studies of the effects of sensory perception on aging and lifespan in invertebrates dates back at least to the work of Apfeld and Kenyon in *Caenorhabditis elegans* (*Apfeld and Kenyon, 1999*), and in the years since, a variety of sensory modalities, including smell, taste, sight, and pain, have become established as important modulators of aging across invertebrate taxa (*Libert et al., 2007*; *Waterson et al., 2015*; *Riera and Dillin, 2016*). Exposure of flies and worms to food-based odorants, for example, limit the benefits of dietary restriction and influence measures of healthy aging, including sleep and daily activity patterns (*Miller et al., 2022*; *Linford et al., 2015*). Some of these studies have focused on how perception of conspecific pheromones, detected through olfaction and gustation, can modulate *Drosophila* and *C. elegans* lifespan (*Shi and Murphy, 2014*; *Maures et al., 2014*; *Gendron et al., 2014*). Although there is no guarantee that mechanisms linking pheromones to aging in worms and flies will prove to be analogous to pathways in mice, they do provide clues that deserve to be explored. Worm perception impacts insulin-specific pathways, and conserved motivation and reward neuropeptides, such as NPF/NPY, are required for aging effects in flies (*Shi and Murphy, 2014*; *Maures et al., 2014*; *Gendron et al., 2014*). It seems reasonable to test whether these same pathways influence mammalian lifespan in mice, either with, or in the absence of, sensory cues of the kind revealed by our experimental findings.

As far as we know, this is the first observation that lifespan can be increased, in a mammal, by olfactory signals, or indeed secreted factors found in soiled bedding and urine. More generally, the work hints that very young mice, and perhaps animals of other species, are able to assess aspects of the social environment and adjust their development to improve their Darwinian fitness and, as a side effect, slow the rate of aging and maintain health in later life.

## Methods

### Mice

We conducted an initial preliminary experiment on changes in sexual maturity with urine and bedding exposure using UM-HET3 mice. The mothers of the UM-HET3 mice used in the preliminary experiment were CByB6F1/J, JAX stock #100009, whose female parents are BALB/cByJ and whose male parents are C57BL/6J. The fathers of the test mice were C3D2F1/J, JAX stock #100004, whose mothers are C3H/HeJ, and whose fathers are DBA/2J.

All mice used in the main lifespan experiment were on a mixed background of the C57BL/6J and S129 mouse strains. They contained a version of the *Gnao1* gene (*Gnao1*) that had been floxed at exons 5 and 6. These were crossed with mice carrying a transgene of Cre recombinase that caused Cre expression under the control of olfactory marker protein (*Omp*cre) (*Li et al., 2004*). Prior research has shown that *Gnao1* expression is strongly reduced in the VNO of *Gnao1* flox/flox/*Omp*cre/+ (cGαo-/-) mice, and Gαo protein immunoreactivity is absent in the VNO sensory epithelium and caudal accessory olfactory bulb (*Chamero et al., 2011*). By contrast, *Gnao1* expression is unaltered in the main olfactory epithelium and in whole brain samples (*Chamero et al., 2011*). Since their initial development and validation, these mice have been maintained as a stock at Saarland University (Germany) and have been used to show that cGαo-/- mice show impairments in aspects of reproductive (*Oboti et al., 2014*) and aggressive behavior (*Chamero et al., 2011*), predator avoidance (*Pérez-Gómez et al., 2015*), in addition to diminished responses to specific bacterial peptides (*Bufe et al., 2019*).

A breeding stock of *Gnao1* flox/flox/*OMP*cre mice were shipped to the University of Michigan (USA) from Saarland University. On arrival mice were genotyped for both *Omp*cre and the floxed *Gnao1* gene as outlined below. Breeding animals were established with parents that were both homozygous for the floxed *Gnao1* gene, with one parent in each pair (either the mother or the father) also containing one copy of the transgene for Cre recombinase. This mating system led to the production of Cre heterozyous mice (*Gnao1*fx/fx, *Omp*cre/+, with deletion of Gαo in VSNs) and Cre negative (*Gnao1*fx/fx, *Omp*+/+, control) mice in equal proportions, with these litters used in the subsequent lifespan experiment. Progeny from this breeding population have also been used in other experiments that show reduced metabolic responses to females and their odors in cGαo-/- male mice (*Garratt et al., 2022*).

### Genotyping

A small tail tip sample was taken from each mouse at weaning, and animals were genotyped for *Omp* and Cre recombinase with the following primers, *Omp* -F: TGGCAACAGCTGTAGCACTT; *Omp*-R: ACAGAGGCCTTTAGGTTGGC; CRE-F: CATTTGGGCCAGCTAAACAT; CRE-R: CCCGGCAAAACAGGTAGTTA. A PCR was conducted with Phire Green Hot Start II PCR Master Mix (Thermo Fisher, Waltham, MA,USA), using the following cycling program (Step 1: 2 min 95°; Step 2: 45 s 94°; Step 3: 30 s 58**°**; Step 4: 45 s 72°; repeat steps 2–4 x38; Step 5: 10 s 72°; Step 6: ∞ 4°). Amplified samples were run on a 1.5% Agar gel with *Omp* producing a 500 bp amplicon and Cre producing a 453 bp amplicon.

Animals were genotyped for the floxed *Gnao1* gene with the following primers, *Gnao1* F: AAGAATAGAACCTAGGACTGGAGG; *Gnao1* R: GCAGACAAGTGAACAAGTGAA ACCC. A PCR was conducted with Phire Green Hot Start II PCR Master Mix (Thermo Fisher, Waltham, MA,USA) using the following cycling program (Step 1: 15 min 95°; Step 2: 40 s 94°; Step 3: 30 s 58**°**; Step 4: 90 s 68°; repeat steps 2–4 x35; Step 5: 420 s 72°; Step 6: ∞. 4°). Amplified samples were run on a 1.4% Agar gel with the WT allele producing a 1868 bp amplicon and the floxed allele producing a 2142 bp amplicon.

### Lifespan experiment design

The proposed lifespan experiment was approved by the University of Michigan Animal Care and Use Program, Protocol PRO00007884. All experiments strictly adhered to this approved protocol. Mice in each individual litter were randomly assigned to a specific treatment arm (male odor, female odor, or

no odor, see below) at age 3 days, and were weaned to same-sex cages at four mice per cage at 19 or 20 days of age. Litters of size 7 or lower were not used, and litters with more than 8 pups were trimmed to 8 pups as soon as discovered. A tail snip biopsy was taken for genotyping at age 10–19 days. Each cage of weanlings contained four mice, two WT and two mutant, and each mouse in a cage was in the same odor-exposure treatment group. Cages were labeled with details of mouse sex, genotype, and treatment group and so were not blinded, although the technician team that assessed mice on a daily basis were largely unaware of the principal hypotheses of the proposed experiment. Thirty mice were allocated per sex, genotype, and odor treatment combination with each mouse assumed to be an independent biological replicate. This sample size was selected because it provides sufficient power to detect a change in median lifespan of approximately 10–15% within one sex and genotype. We originally hypothesized that some effects would be consistent across sexes allowing sexes to be pooled for analysis, further increasing statistical power. All mice received Purina 5LG6 food and water without restriction. Animal housing rooms were maintained on a 12:12 hr light:dark cycle, with relative humidity of 30–70%, and temperature of 21–23°C. Animals were maintained in Specific Pathogen Free Conditions, with sentinel mice included within the colony checked quarterly for infectious agents such as pinworm. All tests were negative through the study period.

## Odor exposure preliminary experiment

The experimental protocol for odor exposure was based on previous studies that had manipulated age at sexual maturity with male and female odors. We used an odor exposure protocol that began before weaning because this had previously been shown to have greater effects on female body weight. We also combined a urine exposure protocol prior to weaning with further exposure to soiled bedding after weaning until mice were 60 days of age, in an effort to maximize the chance of inducing a long-lasting effect.

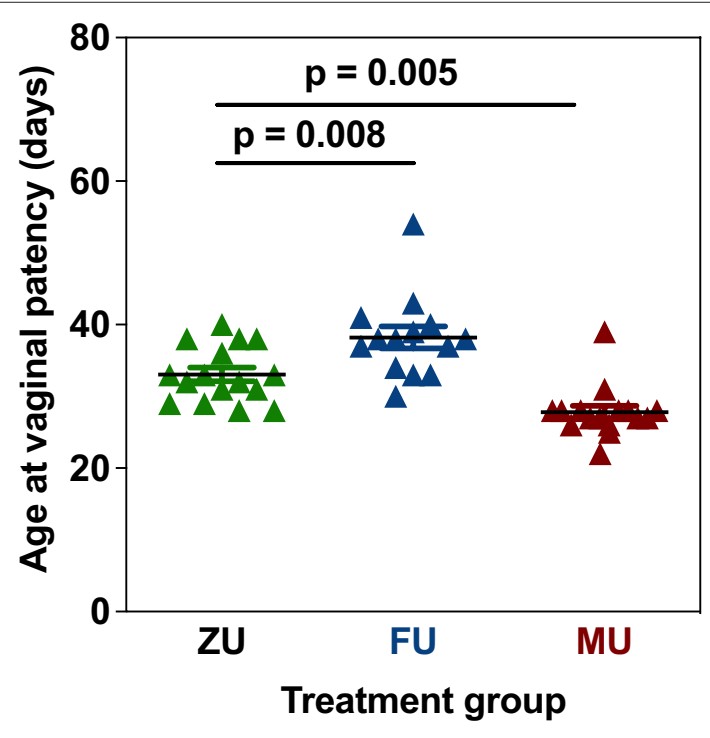

**Figure 4.** Changes in vaginal patency in response to odor treatments in UM-HET3 mice. Each symbol represents the age at vaginal patency in an individual mouse. p Values are from a Sidak post hoc comparison test. Error bars shown the mean ± standard error of the mean (SEM). ZU = control mice, FU = those exposed to female odors, MU those that were exposed to male odors.

The online version of this article includes the following source data for figure 4:

**Source data 1.** Age at vaginal patency for each individual.

We conducted an initial preliminary experiment with this odor exposure protocol using UM-HET3 mice, to test whether this odor protocol would lead to the expected changes in sexual maturity in female mice (delayed with exposure to female odors, hastened in response to male odors), the sex where changes in sexual maturity with odor exposure have been assessed in most prior studies.

Litters of UM-HET3 mice were exposed to either control, male or female odors (6 litters per treatment, n=16 females per treatment for control and male odor exposed mice, n=14 females for female odor exposed mice) according to the protocol outlined below from the day after birth until 60 days of age (see odor exposure experimental protocol). From the date of weaning, one cage of females from each of these litters was visually checked daily for vaginal patency, defined as opening of the vulva, which is an early event that is a predictive index of the first oestrus in mice (*Caligioni, 2009*). This experiment showed that there was a significant difference among the odor treatment groups in vaginal patency. Mice exposed to male odors had an earlier vaginal patency compared to controls, and those exposed to female odors had a later age at vaginal patency (*Figure 4*; p<0.001 for ANOVA, p=0.008 for Sidak post hoc comparison between control and female odor treated mice, p=0.005 for Sidak post hoc comparison between control and male odor-treated mice). This was used as justification of the experimental design and the same treatment protocol was thus applied in the lifespan experiment.

## Odor exposure experimental protocol

Urine was collected from virgin group-housed (three males or four females per cage) young adult (6 months of age) male or female UM-HET3 mice (*Miller et al., 1999*), and stored in aliquots at –20°C until use. Aliquots were then thawed when needed and discarded within 24 hr of thawing; no aliquot was used on more than a single day. Newborn mice had their noses moistened, using a cotton swab, with female urine (FU), male urine (MU), or autoclaved water (ZU, control) 6 days each week, starting at day 3 of life, where day 1 is the date on which pups were first noted in the breeding cage. Urine was applied in the light period between 8 am and noon each day. The mother was placed in a clean cage just prior to odor exposure of her pups, and then returned to the nursing cage immediately thereafter.

After weaning, mice were exposed to same-sex, opposite-sex, or control soiled bedding instead of urine. Spent bedding was obtained from cages housing four adult female or three adult male UM-HET3 mice, aged 4–12 months, after the cage had been occupied for 7 days. Bedding from each donor cage was thoroughly mixed by hand, and then 10% of bedding from a donor cage was placed in each recipient cage, with an equal proportion of the bedding from the recipient's cage removed. Spent bedding was added once each day for 6 days per week until mice were 60 days of age. For control cages, no bedding was added but the bedding in the cage was re-arranged to simulate the MU and FU intervention. After 60 days of age there was no further odor exposure for mice in any of the treatment groups.

## Lifespan data

Lifespan analysis and criteria for inclusion in final survival statistics followed the established protocols of the Interventions Testing Program (*Macchiarini, 2021*). Each mouse was inspected daily and the date of natural death recorded. Mice were euthanized to comply with humane-use protocols if they appeared unlikely to survive more than another 7 days using a symptom checklist. Animals were excluded from analysis and censored if they were injured due to fighting, showed malocclusion, or had dermatitis covering greater than 20% of their body. Six females and two males were removed from the study as a consequence of these exclusion factors, leaving the final sample sizes for survival analysis presented in *Table 1*.

## Assessment of temperature, grip strength, and rotarod performance

Mice were evaluated at 22 months of age. Cages of mice were transferred to the testing room between 8 am and 10 am, and immediately evaluated for core body temperature using a rectal thermometer (Braintree RET3). Mice in the cage were then individually weighed. The mice were then allowed to acclimate in their home cage for 1 hr prior to further testing, after which grip strength was evaluated. Mice were held by the base of the tail and lowered toward a wire grid connected to a strain gauge (Bioseb Research Instruments). Once both forepaws grasped the grid, they were firmly but gently pulled horizontally until they released their grip. Strain gauge measurements were recorded for each

of three trials for each mouse ('forepaw grip'). Three additional trials were then conducted, allowing all four paws to simultaneously grasp the grid ('allpaws grip'). The cage was then returned to its original housing room. On the next day, cages were again moved to the testing room and mice allowed to acclimate for an hour, after which a RotaRod test device (Ugo Basile) was used to measure motor coordination on an accelerating, rotating rod, for three trials. Mice were placed on the RotaRod at an initial speed of 5 RPM. The RotaRod accelerated to a maximum of 40 RPM over 300 s. The trial ended when the mouse fell from the RotaRod, and the latency to fall was recorded. Each mouse was tested three times, with a 1 min interval between trials to allow the apparatus to be cleaned. The mean forepaw grip strength and rotarod performance for each mouse was used for the analysis of genotype and odor effects as previously reported (*Herrera et al., 2020*; *Garratt et al., 2019*).

## Assessment of non-fasting blood glucose and weight

Blood samples (approximately 50 µm) were taken by tail venipuncture from restrained, non-anesthetized 12-month-old mice into 300 µm tubes containing EDTA, and tested using a Bayer Contour Next EZ Glucometer. Body weight was periodically assessed between 9 am and noon.

## Statistical analysis

We tested for survival differences among treatment groups, for each sex, using the log-rank test, with the two genotypes pooled. When the overall log-rank test, with all three odor groups, failed to confirm the null hypothesis, we then proceeded to consider all pairs of odor groups using the log-rank test. To determine if genotype modulated the response to odors, we used Cox regression with two factors, genotype and odor group, including the interaction term, and did so for each sex separately. The significance of the interaction term was used as the indicator of whether response to odor was or was not genotype-dependent within each sex. As a surrogate for 'maximum' lifespan, we used the Wang/Allison test (*Wang et al., 2004*), comparing odor groups for the proportion of mice still alive at the 90th percentile of the joint survival distribution. Calculations of median and 90th percentile age were performed on the number of mice shown in *Table 1*, that is, excluding mice that had died because of fighting or other accident. All non-longevity data were analyzed using an ANOVA. Where an overall treatment group effect was detected we used the Sidak post hoc multiple comparison test to determine whether specific odor treatment groups differed compared to controls. All analyses were conducted using STATA version 17.

## Acknowledgements

This work was supported by NIA grant AG024824 and by the Glenn Foundation for Medical Research. Funding from the American Federation for Aging Research, The Michigan Society of Fellows, The Marsden Fund and Deutsche Forschungsgemeinschaft (DFG) Grants Sonderforschungsbereich SFB 894 and SFB/TRR 152 (to FZ and TL-Z) are also acknowledged.

## Additional information

### Funding

| Funder | Grant reference number | Author |
| --- | --- | --- |
| National Institute for Aging | AG024824 | Richard A Miller |
| Glenn Foundation for Medical Research | | Scott D Pletcher Richard A Miller |
| Deutsche Forschungsgemeinschaft | SFB 894 | Frank Zufall Trese Leinders-Zufall |
| Deutsche Forschungsgemeinschaft | SFB/TRR 152 | Frank Zufall Trese Leinders-Zufall |

The funders had no role in study design, data collection and interpretation, or the decision to submit the work for publication.

## Author contributions
Michael Garratt, Scott D Pletcher, Conceptualization, Funding acquisition, Methodology, Writing - review and editing; Ilkim Erturk, Roxann Alonzo, Investigation; Frank Zufall, Trese Leinders-Zufall, Resources; Richard A Miller, Conceptualization, Resources, Funding acquisition, Investigation, Methodology, Writing - original draft

## Author ORCIDs
Michael Garratt (iD) http://orcid.org/0000-0002-9383-3313
Frank Zufall (iD) http://orcid.org/0000-0002-4383-8618
Trese Leinders-Zufall (iD) http://orcid.org/0000-0002-0678-362X
Scott D Pletcher (iD) http://orcid.org/0000-0002-4812-3785

## Ethics
The proposed lifespan experiment was approved by the University of Michigan Animal Care and Use Program, Protocol PRO00007884. All experiments strictly adhered to this approved protocol.

## Decision letter and Author response
Decision letter https://doi.org/10.7554/eLife.84060.sa1
Author response https://doi.org/10.7554/eLife.84060.sa2

---

# Additional files

## Supplementary files
• MDAR checklist

## Data availability
All data presented in the manuscript is contained within the Supplementary source data file.

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
