## [Editor Report]

This valuable study provides solid evidence for a new intervention, exposure to male vs. female olfactory cues, with an impact on female mouse lifespan. This is interesting to the field of aging research, especially since most described pro-longevity interventions to date tend to work better in male mice.

---

## [Decision Letter]

**Decision letter after peer review:**

Thank you for submitting your article "Lifespan Extension in Female Mice By Early, Transient Exposure to Adult Female Olfactory Cues" for consideration by *eLife*. Your article has been reviewed by 3 peer reviewers, one of whom is a member of our Board of Reviewing Editors, and the evaluation has been overseen by Carlos Isales as the Senior Editor. The following individual involved in the review of your submission has agreed to reveal their identity: Yu-Xuan Lu (Reviewer #3).

Essential revisions:

After discussion, the reviewers agreed that for rigor and reproducibility of the analysis, the authors should:

(i) plot all data, regardless of significance (i.e. both for male vs male, all functional phenotypes tested);

(ii) plot data segregated by genotype, not just pooled;

(iii) address all noted caveats (i.e. cross genotype exposure of urine, potential traces of fecal matter/microbiome in soiled bedding), at least in the discussion;

(iv) provide more information about the Gao KO, including genotyping protocol and validation that the gene was knocked out (essential to conclude that Gao is not involved in the phenotype).

*Reviewer #1 (Recommendations for the authors):*

1. Although the experiments were performed in WT vs. Gnao1 mutant animals, the data is only reported pooled (e.g. Figure 1, Figure 2). Although the authors say that there is no genotype effect, it is crucial that the data presented in Figures 1 and 2 be also provided segregated by genotype, so that it is clear to the readers that no genotype effect (even if potentially underpowered) is present.

2. The authors performed the experiments on a mixed C57BL6/Sv129 background, but urine and soiled bedding were obtained from UM-HET3 mice. Since there may be variation in olfactory cues and receptors between strains, this should be discussed as a caveat for the study.

3. Although methods indicate that all mice survive to 22 months (which should be >50% of the cohort for both females and males), functional data reported in Figure 2 is only so for 1 sex each. The text indicates data not shown for the other sex – this is not acceptable, as all data should be provided if only as a supplemental panel to support all statements. In addition, glucose is reported for males, which did not show survival benefits. However, we suggest plotting both female and male data for the 3 parameters for easy comparison by the readers. It would also be beneficial to color code by genotype. The data from the rotarod and grip strength test, although not significant, should also be plotted and reported.

4. Please provide all necessary information for how the mice were genotypes (i.e. DNA extraction from the tail, PCR master mix and cycling conditions, primer sequences).

*Reviewer #2 (Recommendations for the authors):*

– Spent bedding was added for 12 weeks. Does that mean after 12 weeks, all mice were placed on standard bedding with no disruption?

– What was the temperature, humidity, and light/dark cycle for the animal facility?

– What statistics did you use for any of the non-longevity data? You state ANOVA in your results, but it should also be in your methods. Also for rotarod, you need to control for body weight.

– Is there a reason you did glucose measurement at 12 months, but all other health measures at 22 months?

– In your body temperature analysis, it seems like you have two outliers in your ZU group (one definitely outside the normal range of mouse healthy physiology). If you remove these individuals, do you still find a significant difference? I would assume those two very low temp animals were either sick or there was a misreading of the thermometer.

– I think Figure 2 is confusing. It would make more sense to show each variable for both sexes, not just cherry-picking the significant ones. I would also like to see the body weight trajectory curves since they were measured at multiple time points.

– I would include the coefficients and test statistics in Table 2.

– While you found no effect of genotype, I think you need to include what the sample sizes were for each group, not the genotypes pooled in Table 1.

– Did you confirm that the Gao gene was knocked down/out in your mice?

– I'm not sure why the discussion focuses so much on GH when it does not really affect any of the results here, other than as evidence that early life effects can influence lifespan.

*Reviewer #3 (Recommendations for the authors):*

1) In figure 1, the authors did a log-rank text to compare all three treatment groups, and then did follow-up log-rank tests to compare differences between pairs. It would be crucial to run the second test with correction for multiple comparisons and mention in the method section clearly.

2) In figure 2, it would be better to visualise all data points and median. Box plots or violin plots could be good options.

3) In the last sentence of the Abstract, the authors stated "…have long-lasting effects on disease resistance, and…". But the entire study did not perform any direct challenges on testing disease resistance, for instance, infection, etc. I would suggest using "health maintenance at late life" or something similar instead.

---

## [Author Response]

Essential revisions:After discussion, the reviewers agreed that for rigor and reproducibility of the analysis, the authors should:(i) plot all data, regardless of significance (i.e. both for male vs male, all functional phenotypes tested);

We have added several new figure panels in response to this request, showing temperature in both sexes (Figure 2), glucose in both sexes (Figure 2), and weight at all ages in both sexes (Figure 3). We have also added a supplementary figure showing the data for grip strength and rotarod performance (Figure 2—figure supplement 1). All data is now provided in plots.

(ii) plot data segregated by genotype, not just pooled;

Figures 1 and 2 now shows each genotype separately. The supplementary data for grip strength, rotarod performance and body weight (Figure 3—figure supplement 1) are also coded to show the different genotypes. Thus all data is available showing the separate genotypes.

(iii) address all noted caveats (i.e. cross genotype exposure of urine, potential traces of fecal matter/microbiome in soiled bedding), at least in the discussion;

We have added more information about these caveats to the discussion:

“We should note that the genotype of the mice exposed to odor was not the same as the genotype (UM‑HET3 genetically heterogeneous stock) from which urine and bedding was collected. The use of a genetically heterogeneous mouse model as odor donors would mean that the odor receiving animals would be exposed to signaling proteins and volatile odorants produced by several strains of mice. We perceived this as a strength because different mouse strains produce different olfactory signaling proteins [28], and exposing mice to odors from a variety of genotypes would increase the probability of exposure to an odorant that influences lifespan. However, different mouse strains also express different olfactory receptors [29], and could show different physiological responses to bedding from different strains, although VNO activation responses in mice to odors from the same or different strain of mouse has been reported as broadly similar [30, 31]. In principle, it’s possible that different lifespan responses may occur if mice are exposed to odors from a background genotype.”

“As our results were not significantly altered by the olfactory genetic manipulation employed in this experiment, it is possible that a different sensory modality might be involved in mediating these effects. Similarly, we cannot exclude the possibility that microbiota transferred with the bedding, rather than the urine odor and bedding odor per se, cause these effects. Regardless, this must be a female-specific secreted factor that only influences female lifespan, and can do so even when females are exposed only over the first 12 weeks of life”.

(iv) provide more information about the Gao KO, including genotyping protocol and validation that the gene was knocked out (essential to conclude that Gao is not involved in the phenotype).

We have now provided a complete description of the genotyping protocol. We have also provided more information on the validation of this model from previous studies, including specific information on conditional loss of Gao in the vomeronasal organ and assessory olfactory bulb but not the main olfactory epithelium. This deletion was comprehensively described in the original study by coauthors Frank Zufall and Trese Leinders-Zufall, documenting impaired aggressive behavior in cGao -/- mice. Since then, this mouse model has also been used to show that cGao -/- mice exhibit impaired reproductive behavior (including sexual maturity) and a reduced ability to detect some bacterial peptides. We also have another preprint currently under revision that shows male cGao -/- mice show impaired metabolic responses to females and their odours. This comprehensive validation and phenotyping is now referred to in the revised article with associated references.

Reviewer #1 (Recommendations for the authors):1. Although the experiments were performed in WT vs. Gnao1 mutant animals, the data is only reported pooled (e.g. Figure 1, Figure 2). Although the authors say that there is no genotype effect, it is crucial that the data presented in Figures 1 and 2 be also provided segregated by genotype, so that it is clear to the readers that no genotype effect (even if potentially underpowered) is present.

In Figure 1, we have now presented the survival data for each of the two genotypes, in addition to the original presentation in which the data were pooled across genotype. In Figure 2, we have used color coding to distinguish individual mice of the two genotypes.

2. The authors performed the experiments on a mixed C57BL6/Sv129 background, but urine and soiled bedding were obtained from UM-HET3 mice. Since there may be variation in olfactory cues and receptors between strains, this should be discussed as a caveat for the study.

As noted above in our reply to the editor, we have added a paragraph to the discussion to note these two ideas.

3. Although methods indicate that all mice survive to 22 months (which should be >50% of the cohort for both females and males), functional data reported in Figure 2 is only so for 1 sex each. The text indicates data not shown for the other sex – this is not acceptable, as all data should be provided if only as a supplemental panel to support all statements. In addition, glucose is reported for males, which did not show survival benefits. However, we suggest plotting both female and male data for the 3 parameters for easy comparison by the readers. It would also be beneficial to color code by genotype. The data from the rotarod and grip strength test, although not significant, should also be plotted and reported.

We have modified Figure 2 to show both sexes, as requested, and to indicate the genotype of each mouse tested.

We have added plots of mean forepaw grip strength and rotarod performance as supplementary information. None of these showed any effect of odor exposure or genotype.

4. Please provide all necessary information for how the mice were genotypes (i.e. DNA extraction from the tail, PCR master mix and cycling conditions, primer sequences).

This is now provided in the methods.

Reviewer #2 (Recommendations for the authors):– Spent bedding was added for 12 weeks. Does that mean after 12 weeks, all mice were placed on standard bedding with no disruption?

Yes. We have now clarified this in the methods.

– What was the temperature, humidity, and light/dark cycle for the animal facility?

We have added information on temperature, humidity and light cycle to methods.

– What statistics did you use for any of the non-longevity data? You state ANOVA in your results, but it should also be in your methods. Also for rotarod, you need to control for body weight.

We have added reference to using an ANOVA for non-longevity data in the methods, and the associated post-hoc test. Given that there is no differences between the groups for rotarod function or body weight, and we have measured multiple parameters, we feel that it is not worth including the body weight controlled data. Just to check, for grip strength and rotarod performance we also reran each ANOVA including body weight as a covariate. This analysis also showed no effect of genotype or treatment for these parameters.

– Is there a reason you did glucose measurement at 12 months, but all other health measures at 22 months?

We measure temp, grip, and rotarod at 22 months because these vary with age, and we wanted to look for possible delay in age changes. We do not like to take blood samples at that age, because of a concern that this might diminish survival for some mice, so we took the glucose sample at mid-life. Glucose was not being used as an age-sensitive variable, but as a possible mediating factor in the process by which early life odors alter lifespan.

– In your body temperature analysis, it seems like you have two outliers in your ZU group (one definitely outside the normal range of mouse healthy physiology). If you remove these individuals, do you still find a significant difference? I would assume those two very low temp animals were either sick or there was a misreading of the thermometer.

If we ignore the lowest temperature, we obtain p = 0.03 for the overall ANOVA and p = 0.03 for the comparison of ZU to FU. If we ignore the two lowest temperatures, we obtain p = 0.07 for the ANOVA and p = 0.11 for the post-hoc comparison of FU to ZU. You may be right that one or both of these animals might have been sick, or (less likely) that the test was performed incorrectly, but we do not have any evidence to support or refute those ideas. Our preference is not to eliminate data points without a good reason, observational or statistical, to do so.

– I think Figure 2 is confusing. It would make more sense to show each variable for both sexes, not just cherry-picking the significant ones. I would also like to see the body weight trajectory curves since they were measured at multiple time points.

We have added the data suggested by this reviewer.

– I would include the coefficients and test statistics in Table 2.

STATA’s Cox regression algorithm calculates hazard ratios, standard errors, z-scores, and 95% confidence intervals comparing each group to its baseline. It then computes p-value and Χ2 statistics for the marginal effect of each factor (odor, genotype and interaction); it is these marginal p-values that were tabulated in Table 2. We have now added the Χ2 value to address the reviewer’s request.

– While you found no effect of genotype, I think you need to include what the sample sizes were for each group, not the genotypes pooled in Table 1.

Groups were very nearly equal in size, with the number of WT mice ranging from 27 to 30, and the number of mutant mice ranging from 28 to 30. We have added the count of WT mice in each class to Table 1 as requested.

– Did you confirm that the Gao gene was knocked down/out in your mice?

The phenotype of these animals has been comprehensively described in previous studies, showing the loss of Gao in basal vomeronasal sensory neurons. As highlighted above the knockout of Gao was well characterized in the first study that generated these mice, and then three subsequent studies have demonstrated that this knockout impairs aspects of reproductive behavior, predator avoidance and detection of bacterial peptides.

– I'm not sure why the discussion focuses so much on GH when it does not really affect any of the results here, other than as evidence that early life effects can influence lifespan.

Our key conclusions are (a) that transient early life environment can lead to enduring changes in state sufficient to extend lifespan, and (b) that odor can cause such a shift. Much of the prior evidence on transient early life environmental changes comes from manipulations of GH, and thus a synopsis of these experiments provides a useful context for the reader. The GH-injected design also points to possible mechanisms, including changes in stress resistance, hypothalamic inflammation, and the set of cellular changes listed in Ref [6]; listing these downstream phenotypes provides suggestions for follow-up work in our new odor exposure model

Reviewer #3 (Recommendations for the authors):1) In figure 1, the authors did a log-rank text to compare all three treatment groups, and then did follow-up log-rank tests to compare differences between pairs. It would be crucial to run the second test with correction for multiple comparisons and mention in the method section clearly.

The overall log-rank test, significant at p = 0.01, provides protection against multiple comparison errors, but a Bonferroni corrected p-value for the post-hoc tests is still statistically significant at p = 0.04. We have added this information to the text.

2) In figure 2, it would be better to visualise all data points and median. Box plots or violin plots could be good options.

Figure 2, both in original and in its revised and expanded version, does visualize all individual plots. We have now adjusted the spread of the datapoints so that each individual point can be more clearly seen. We have also added a mean line to each plot.

3) In the last sentence of the Abstract, the authors stated "…have long-lasting effects on disease resistance, and…". But the entire study did not perform any direct challenges on testing disease resistance, for instance, infection, etc. I would suggest using "health maintenance at late life" or something similar instead.

We have adjusted this sentence so that it refers to maintaining health in later life.